# Requirements for RNA polymerase II preinitiation complex formation in vivo

**Natalia Petrenko[1†], Yi Jin[1†], Liguo Dong[2], Koon Ho Wong[3]\*, Kevin Struhl[1]\***

[1]Department of Biological Chemistry and Molecular Pharmacology, Harvard Medical School, Boston, United States; [2]Faculty of Health Sciences, University of Macau, Macau, China; [3]Institute of Translational Medicine, University of Macau, Macau, China

**Abstract** Transcription by RNA polymerase II requires assembly of a preinitiation complex (PIC) composed of general transcription factors (GTFs) bound at the promoter. In vitro, some GTFs are essential for transcription, whereas others are not required under certain conditions. PICs are stable in the absence of nucleotide triphosphates, and subsets of GTFs can form partial PICs. By depleting individual GTFs in yeast cells, we show that all GTFs are essential for TBP binding and transcription, suggesting that partial PICs do not exist at appreciable levels in vivo. Depletion of FACT, a histone chaperone that travels with elongating Pol II, strongly reduces PIC formation and transcription. In contrast, TBP-associated factors (TAFs) contribute to transcription of most genes, but TAF-independent transcription occurs at substantial levels, preferentially at promoters containing TATA elements. PICs are absent in cells deprived of uracil, and presumably UTP, suggesting that transcriptionally inactive PICs are removed from promoters in vivo.
DOI: https://doi.org/10.7554/eLife.43654.001

**\*For correspondence:**
KoonHoWong@umac.mo (KHW);
kevin@hms.harvard.edu (KS)

†These authors contributed
equally to this work

**Competing interest:** See
page 16

**Reviewing editor:** Michael R
Green, Howard Hughes Medical
Institute, University of
Massachusetts Medical School,
United States

## Introduction

Transcription by RNA polymerase (Pol) II requires assembly of a preinitiation complex (PIC) composed of general transcription factors (GTFs) bound at the core promoter (*Conaway and Conaway, 1993*; *Buratowski, 1994*; *Orphanides et al., 1996*; *Roeder, 1996*). However, despite considerable work over the past three decades, it remains unclear which GTFs are absolutely required for PIC formation and Pol II transcription.

GTFs were defined originally as factors necessary and sufficient for 'basal' transcription from core promoters in vitro. However, such in vitro reactions varied with respect to the promoter used, the concentration and purity of GTFs, and the concentration and nature (e.g. supercoiled, circular, or linear) of the DNA template. Aside from Pol II itself, GTFs include the TATA-binding protein (TBP), TFIIA, TFIIB, TFIIE, TFIIF, and TFIIH. Under some conditions, removal (to the extent possible) of any one factor from the complete reaction causes a drastic reduction in transcriptional activity. However, TFIIA is dispensable or only mildly stimulatory under other conditions (*Ozer et al., 1994*; *Sun et al., 1994*; *Yokomori et al., 1994*). TFIIE, TFIIF, and TFIIH can be dispensable or only stimulatory for transcription from negatively-supercoiled templates, depending on the promoter (*Parvin and Sharp, 1993*; *Goodrich and Tjian, 1994*; *Parvin et al., 1994*; *Timmers, 1994*). TBP-independent transcription that requires only YY1, TFIIB, and Pol II has been observed on supercoiled templates with the YYI initiation element (*Usheva and Shenk, 1994*). In addition to the classically defined GTFs, TBP-associated factors (TAFs) in the TFIID complex and the Mediator complex are important or required for Pol II transcription in vitro, depending on the promoter and conditions (*Tjian and Maniatis, 1994*; *Takagi and Kornberg, 2006*).

In vitro, the PIC is a stable entity that initiates transcription upon addition of nucleotide triphosphates (*Conaway and Conaway, 1993*; *Buratowski, 1994*; *Orphanides et al., 1996*; *Roeder, 1996*).

PIC assembly is initiated with the binding of TBP (or the TFIID complex) to the promoter, as none of the other factors can stably bind DNA on their own (*Buratowski et al., 1989*). GTFs can be sequentially added to give a series of stable, 'partial PICs' prior to assembly of a transcriptionally active PIC (*Buratowski et al., 1989*). Such partial PICs are highly informative on the nature of the interactions between GTFs and the promoter, and structures of functional PICs have now been determined at the atomic level (*Sainsbury et al., 2015*; *Robinson et al., 2016*; *Hantsche and Cramer, 2017*). Functional PICs can vary with respect to the presence or absence of TAFs, Mediator, and TFIIA.

In vivo studies of the role of GTFs for PIC formation and transcription are incomplete. In addition, the requirement for every GTF to permit cell growth makes it impossible to completely eliminate GTF function/activity and to exclude the possibility of indirect effects. Depletion studies indicate that TBP, TFIIB, and Pol II are essential for transcription (*Moqtaderi et al., 1996*; *Fan et al., 2010*; *Wong et al., 2014*), whereas TFIIA is important but not essential (*Chou et al., 1999*; *Liu et al., 1999*; *Stargell et al., 2000*). The kinase subunit of TFIIH (Kin28) is important for promoter escape and Mediator dissociation, but considerable transcription occurs upon its depletion or inactivation (*Jeronimo and Robert, 2014*; *Wong et al., 2014*). Depletion of the entire Mediator complex abolishes Pol II transcription, but Mediator sub-modules can support transcription, albeit at lower levels than the wild-type strain, and can inhibit promoter escape (*Petrenko et al., 2017*). Depletion of Taf1 has led to conflicting results. Most studies indicate a selective role at TATA-less promoters (*Moqtaderi et al., 1996*; *Kuras et al., 2000*; *Li et al., 2002*), whereas others suggest a general requirement for Pol II transcription (*Warfield et al., 2017*).

The relative occupancies of GTFs at promoters are consistent across all promoters (*Kuras et al., 2000*; *Pokholok et al., 2002*; *Rhee and Pugh, 2012*), strongly suggesting that a structurally similar PIC mediates a given level of transcription. Mediator occupancy at core promoters is transient, due to Kin28-dependent dissociation, but it is strongly correlated with GTF occupancies upon Kin28 depletion (*Jeronimo and Robert, 2014*; *Wong et al., 2014*). Nevertheless, while Mediator stimulates PIC formation, it is not an obligate component of the PIC in vivo (*Petrenko et al., 2017*).

TAF occupancy does not strictly correlate with GTF occupancy, providing strong evidence that transcription can be mediated by TAF-containing (i.e. TFIID) and TAF-lacking forms of transcriptionally active TBP, with the relative usage of these two forms depending on the promoter (*Kuras et al., 2000*; *Li et al., 2000*). Depending on which TBP form predominates, promoters can be classified roughly as either 1) constitutive, TATA-lacking, and TFIID-dependent or 2) inducible, TATA-containing, TFIID-independent, and SAGA-dependent (*Struhl, 1986*; *Chen and Struhl, 1988*; *Struhl, 1987*; *Iyer and Struhl, 1995*; *Moqtaderi et al., 1996*; *Basehoar et al., 2004*; *Huisinga and Pugh, 2004*). A variety of other experiments strongly support the idea of functionally distinct forms of TBP. First, the TFIID form is specifically recruited by the Rap1-containing activator and associated NuA4 histone acetylase complex to promoters of ribosomal protein genes (*Li et al., 2002*; *Mencía et al., 2002*; *Uprety et al., 2012*; *Uprety et al., 2015*). In contrast, many other activators do not directly recruit TFIID (*Kuras et al., 2000*; *Li et al., 2000*) but rather recruit the SAGA histone acetylase complex (*Bhaumik and Green, 2001*; *Bhaumik et al., 2004*). Second, TBP is preferentially retained at the TAF-containing vs TAF-lacking promoters upon thermal inactivation of TFIIB or Mediator (*Li et al., 2000*). Third, the relative use of TFIID- vs. SAGA-dependent mechanisms at a given promoter can differ depending on the environmental conditions (*Ferdoush et al., 2018*). However, this view has been challenged by experiments claiming that the two classes of promoters behave similarly upon TFIID depletion (*Warfield et al., 2017*).

Here, we perform a comprehensive analysis of GTF function in vivo by using the anchor-away technique (*Haruki et al., 2008*) to individually deplete each GTF from the nucleus. We demonstrate that all classically defined GTFs are required for PIC formation/stability and Pol II transcription in vivo, suggesting that PICs contain all GTFs. In apparent contrast to some previous observations (*Li et al., 2000*; *Zanton and Pugh, 2006*), our results suggest that partial PICs are unlikely to be stable in vivo. In contrast, while TAFs contribute to Pol II transcriptional activity at most (and perhaps all) genes, we provide direct evidence that TAF-independent transcription occurs at a substantial level. Lastly, we discover that PICs are not observed in cells depleted for uracil, suggesting a mechanism that removes transcriptionally inactive PICs from promoters. Our findings lead to a revised view of the preinitiation complex in vivo.

## Results

### Efficient and rapid depletion of GTFs in vivo

To systematically study the role of GTFs and Pol II on transcription, we constructed anchor-away strains for subunits of TBP (Spt15), TFIIA (Toa1 and Toa2), TFIIB (Sua7), TFIIE (Tfa1 and Tfa2), TFIIF (Tfg1), TFIIH (Ssl1 and Ssl2), and Pol II (Rpb1). In the absence of rapamycin, growth of these strains is comparable to that of an untagged parental strain (*Figure 1A*), indicating that the fusion of the FRB domain to the targeted factors does not significantly affect their function. In contrast, when these proteins are removed from the nucleus by treatment with rapamycin, these strains fail to grow (*Figure 1A*), as expected from the essential roles of GTFs. Binding of the targeted GTFs at active promoters (*PMA1* and *CCW12*) is reduced to background or near-background levels after rapamycin treatment for 1 hr (*Figure 1B*), indicating that depletion of GTFs is highly effective.

### All GTFs are required for pol II transcription in vivo

To examine the effect of depleting individual GTFs on Pol II transcription, we first measured Pol II occupancy at the coding regions of several well-expressed genes. While the addition of rapamycin has minimal effects on transcription in an untagged parental control strain, Pol II occupancy at coding regions of all genes tested is reduced to very low levels upon depletion of any GTF (*Figure 1C* and *Figure 1—figure supplement 1*). To extend these results to genome scale, we performed Pol II ChIP-seq analysis on the same samples to which a known amount of *S. pombe* chromatin was added as an internal control for immunoprecipitation and data normalization. In all cases, depletion of any GTF drastically reduced transcription to near-background levels as determined by metagene (*Figure 2A*) or individual gene (*Figure 2B*) analyses. In contrast, as will be discussed later, depletion of Taf1 results in a modest decrease in transcription. Furthermore, upon TBP depletion, TBP and Pol II occupancies decrease in a kinetically similar manner (*Figure 2C*), indicating that loss of TBP results in an immediate cessation of transcriptional initiation.

In the above experiments, genes are expressed at steady-state levels prior to depletion of the GTF. To address the effect of GTF depletion on inducible transcription, we first depleted cells of an individual GTF and then analyzed the rapid transcriptional activation response to heat shock. In accord with drastic transcriptional effects described on non-inducible genes, induction of *HSP12* (*Figure 3A*) and other heat shock genes (*Figure 3—figure supplement 1A*) is very strongly decreased, although not completely eliminated, for all GTFs (but not Taf1).

The residual levels of transcription observed upon GTF depletion could reflect GTF-independent transcription or, more simply, incomplete depletion of the GTF. In this regard, the anchor-away system works somewhat less efficiently in stress conditions (*Petrenko et al., 2017*). As discussed and shown elsewhere for TBP (*Petrenko et al., 2017*), incomplete depletion of an essential GTF will reduce (but not eliminate) transcription and its occupancy at the promoter, but it will not affect the nature of the PIC and hence the GTF:Pol II occupancy ratio. Indeed, the GTF:Pol II occupancy ratios for all cases of GTF depletion are comparable to that observed prior to depletion (*Figure 3B* and *Figure 3—figure supplement 1B*). Thus, the residual levels of transcription are due to incomplete depletion, indicating that all GTFs are required for Pol II transcription in vivo.

### Depletion of any GTF prevents association of TBP with the promoter, suggesting that partial PICs do not exist at appreciable levels in vivo

As GTFs are components of the PIC, it is expected that their essential role in Pol II transcription reflects their requirement for a functional PIC. However, as partial PICs lacking various GTFs are stable in vitro (*Buratowski et al., 1989*), it is unclear whether such partial PICs are stable in vivo. To address this issue, we examined the effect of GTF depletion on PIC levels by measuring TBP occupancy at the promoter. For all GTFs, TBP occupancy levels at both continually expressed (*Figure 4A* and *Figure 4—figure supplement 1A*) and heat-shock inducible (*Figure 4B* and *Figure 4—figure supplement 1B*) genes are drastically reduced upon depletion. In addition, the TBP:Pol II and TBP:GTF occupancy ratios for all cases of GTF depletion are comparable to that observed prior to depletion and in the parental strain (*Figure 4C* and *Figure 4—figure supplement 1C*). Thus, each GTF is required for a stable PIC in vivo. As TBP is the only GTF that can independently and stably bind to

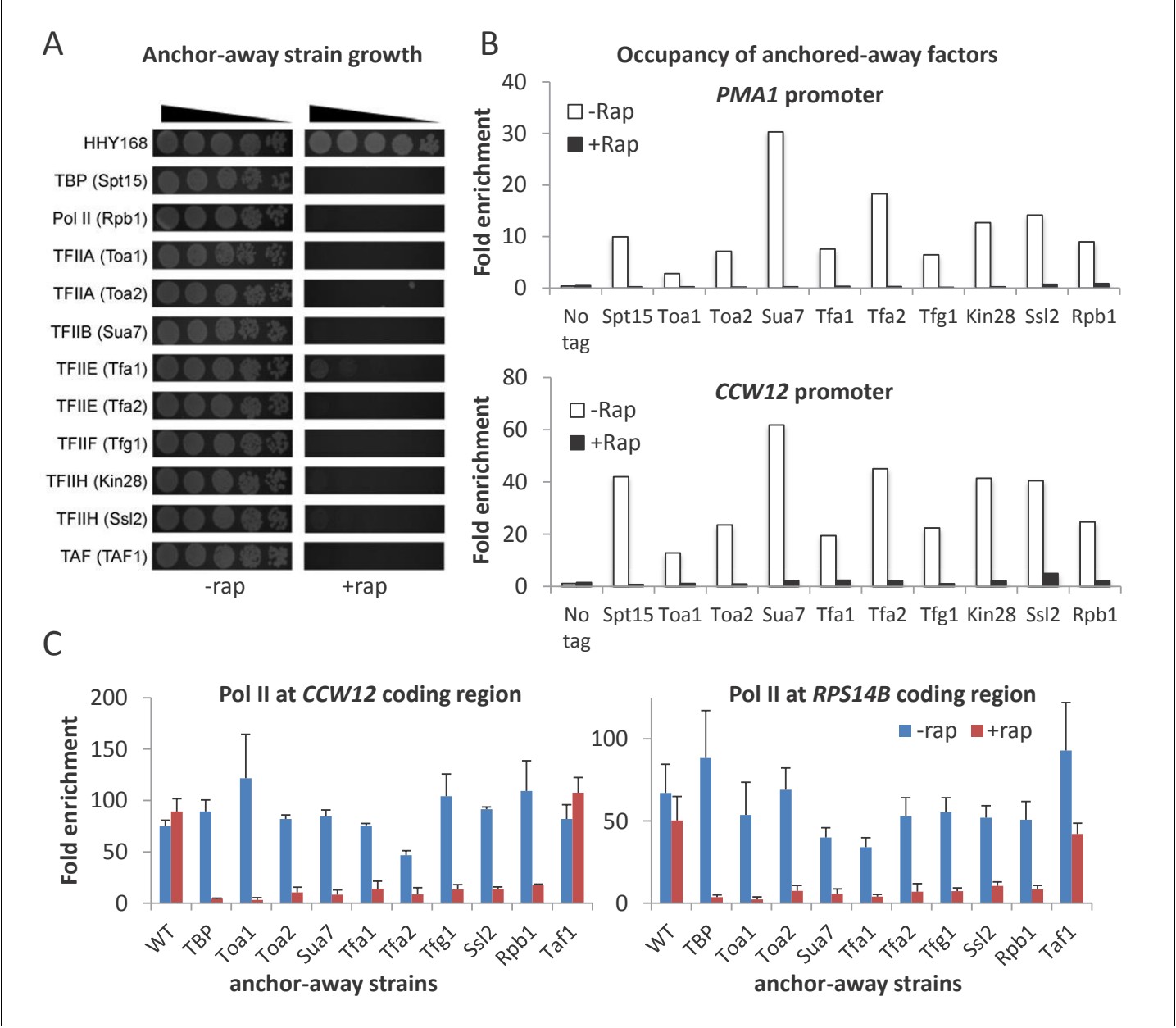

**Figure 1.** Conditional depletion of GTFs causes severe growth and transcriptional defects. (A) Growth of the indicated anchor-away cells (5-fold serial dilutions) in the presence or absence of rapamycin. (B) Occupancy of the indicated FRB-tagged GTFs at the *PMA1* and *CCW12* promoters in the corresponding strains and an untagged control strain in the presence of absence of rapamycin. (C) Pol II occupancy at the *CCW12* and *RPS14B* coding regions in the indicated strains grown in the presence or absence of rapamycin. Error bars represent the standard error of at least three independent experiments.

DOI: https://doi.org/10.7554/eLife.43654.002

The following figure supplement is available for figure 1:

**Figure supplement 1.** Conditional depletion of GTFs causes severe growth and transcriptional defects.

DOI: https://doi.org/10.7554/eLife.43654.003

DNA (*Buratowski et al., 1989*), these results indicate that, unlike the situation in vitro, partial PICs containing GTF subsets are very unstable in vivo.

Our results are in apparent contrast to a previous report claiming the existence of partial PICs in response to a mild heat shock (37°C) based on altered GTF:GTF occupancy ratios (*Zanton and Pugh, 2006*). However, in that report, the altered occupancy ratios for the vast majority of the genes

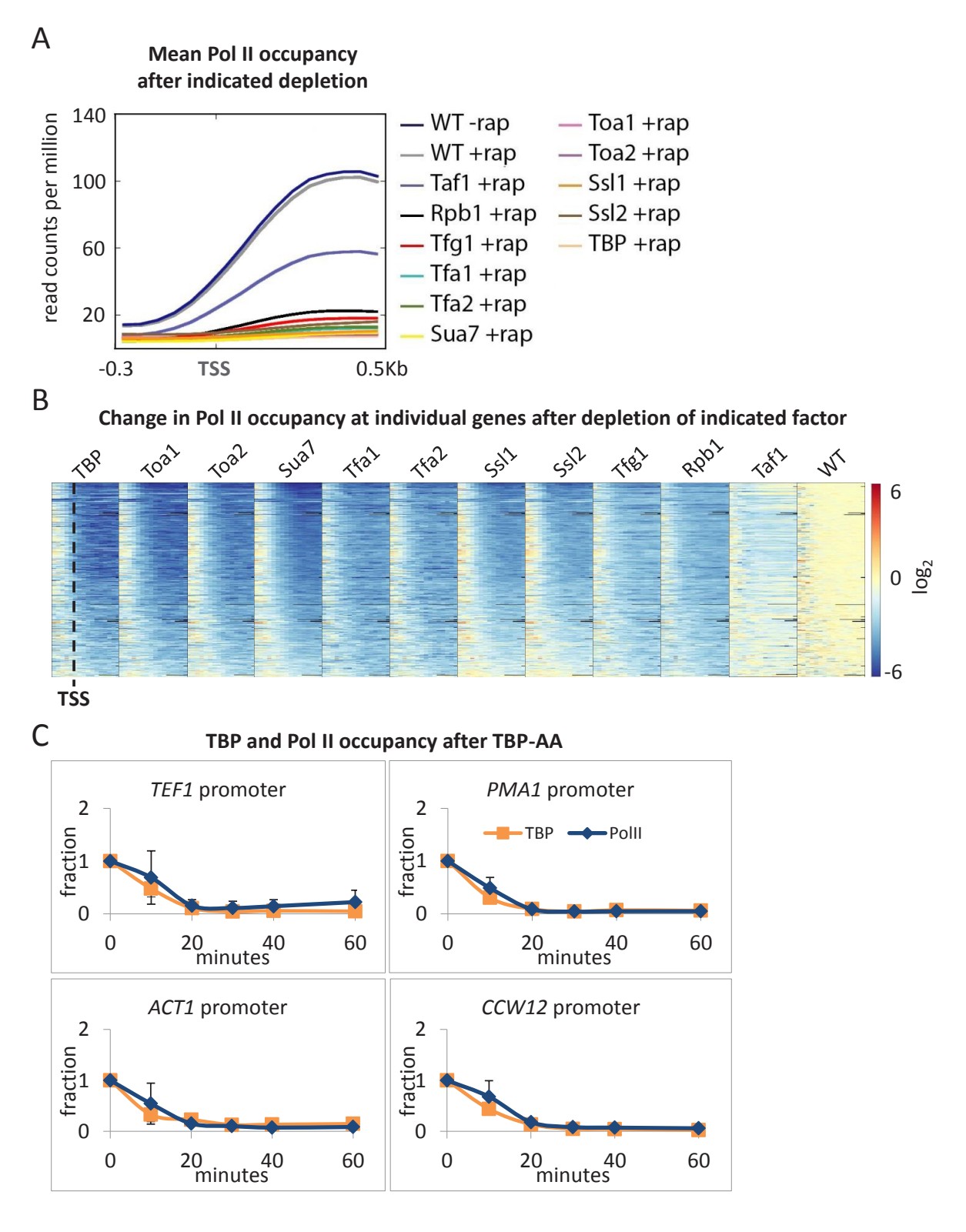

**Figure 2.** All GTFs are generally required for ongoing Pol II transcription. (**A**) Mean Pol II occupancy averaged over 453 well-transcribed genes (metagene analysis) in strains depleted (+rap) for the indicated factor and in the parental (WT) strain (±rap). Partial reduction is observed only for the TAF1-depleted strain. (**B**) Pol II occupancy at individual genes (the same set of 453 genes ordered from top to bottom by expression level in WT) in strains depleted for the indicated factor. For each gene, the log₂ change in Pol II occupancy after depletion is indicated according to the red/blue

*Figure 2 continued on next page*

*Figure 2 continued*

scale. (**C**) TBP and Pol II occupancies at the indicated promoters in the TBP-depletion strain at various times after rapamycin treatment. Error bars represent the standard error of at least three independent experiments.

DOI: https://doi.org/10.7554/eLife.43654.004

with apparent partial PICs are very modest. To address this more directly, we measured the TFIIB and Pol II occupancies at several genes including those analyzed by *Zanton and Pugh (2006)*. We did not observe increased TFIIB:Pol II occupancy ratios in response to heat shock (39°C) at any of these genes (*Figure 4D*). Thus, our results suggest that partial PICs do not exist at appreciable levels in vivo, although their existence cannot be completely excluded.

### The PIC is not stable in the cells depleted of uracil

In vitro, the PIC is extremely stable in the absence of nucleotide triphosphates, and indeed is defined by its ability to initiate transcription upon addition of these precursors. We attempted to mimic this situation in vivo by analyzing PIC levels and transcription under conditions in which *ura3* mutant cells were starved of uracil. As removal of uracil from the medium does not immediately eliminate intracellular uracil (because *ura3* cells require and hence contain uracil for growth prior to the removal), we examined TBP and Pol II occupancy levels at various times after removal of uracil (*Figure 5* and *Figure 5—figure supplement 1A–C*). As uracil depletion causes metabolic mayhem (*Brauer et al., 2008*), we examined genes that are typically inhibited (*Figure 5A* and *Figure 5—figure supplement 1B*), induced (*Figure 5B* and *Figure 5—figure supplement 1C*), or unaffected (*Figure 5C*) by metabolic stress.

Upon removal of uracil from the medium, cells grow at near-normal rates for about 2 hr (using up the intracellular uracil) and then show decreased growth with cessation at about 4 hr (*Figure 5—figure supplement 1A*). With the exception of heat shock genes, TBP and Pol II occupancies at all promoters tested decrease over time, and PIC and transcription (Pol II occupancy at coding regions)

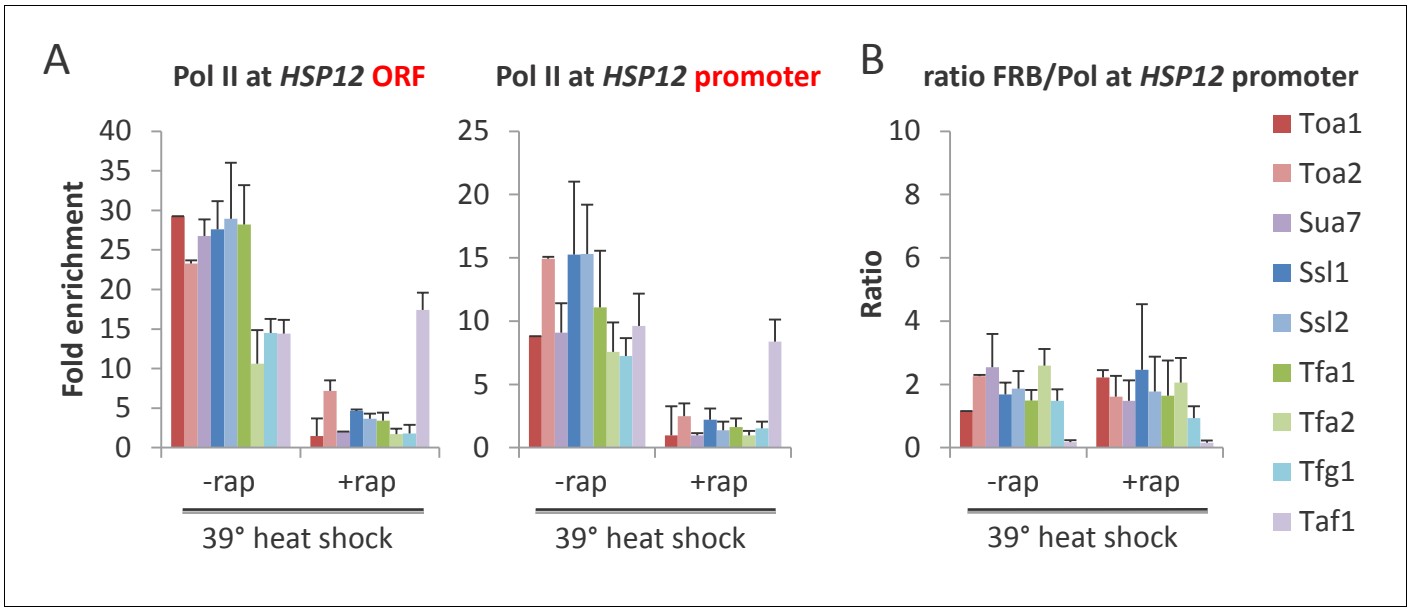

**Figure 3.** All GTFs are required for transcriptional induction upon heat shock. (**A**) Mean Pol II occupancy at the *HSP12* coding region (ORF) and promoter in strains depleted (or not) for the indicated factor and then induced for 15 min by shifting to 39°C. (**B**) FRB-tagged GTF:Pol II occupancy ratio at the induced *HSP12* promoter in cells pretreated or not with rapamycin to deplete the indicated factors.

DOI: https://doi.org/10.7554/eLife.43654.005

The following figure supplement is available for figure 3:

**Figure supplement 1.** All GTFs are required for transcriptional induction upon heat shock.

DOI: https://doi.org/10.7554/eLife.43654.006

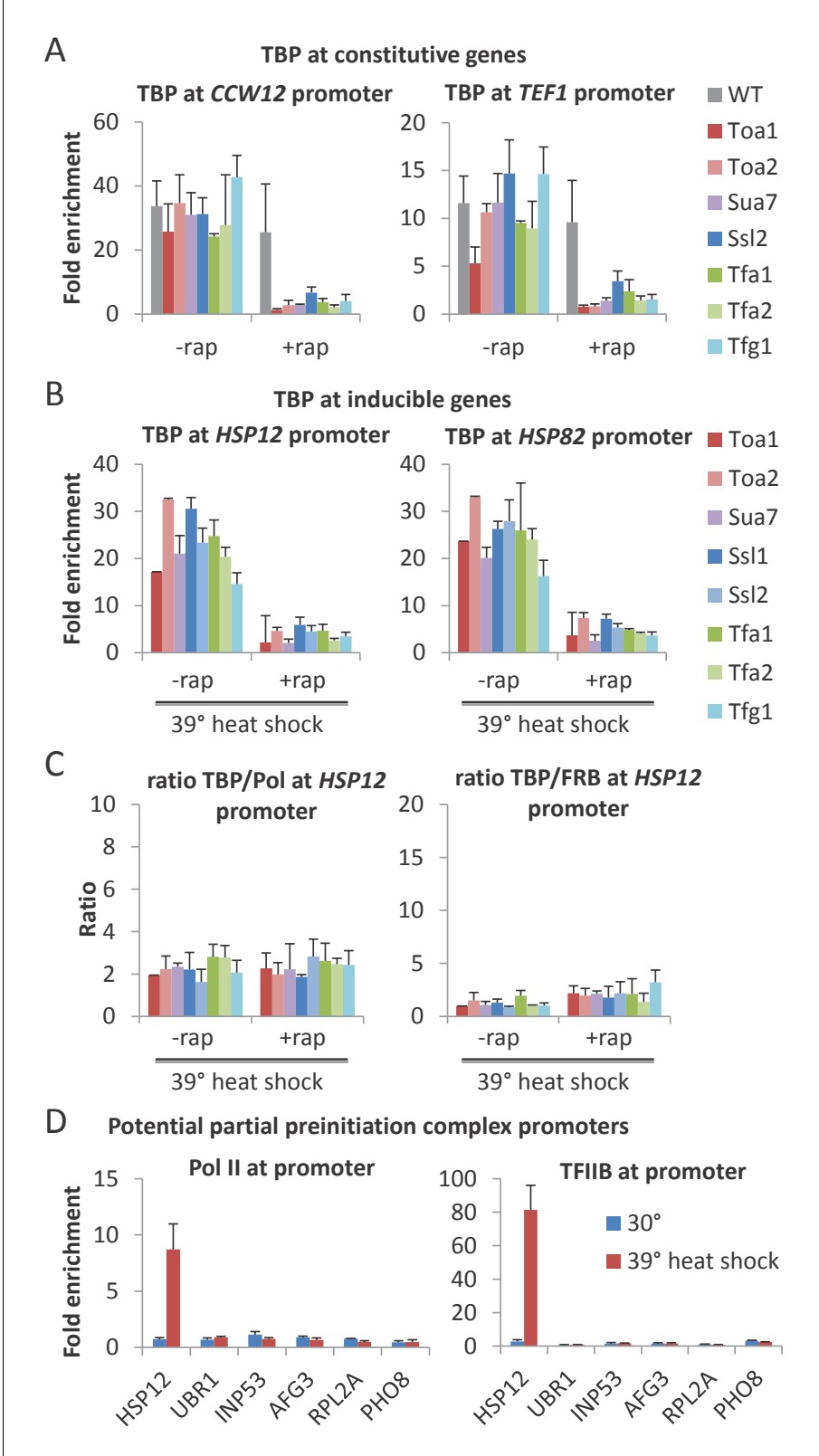

**Figure 4.** All GTFs are required for TBP occupancy, and hence PIC stability/formation. (**A**) TBP occupancy at the *CCW12* and *TEF1* promoters in strains depleted (+rap) or not (-rap) for the indicated factor. (**B**) TBP occupancy at the *HSP12* and *HSP82* promoters in strains depleted (+rap) or not (-rap) for the indicated factor and subject to a heat shock. (**C**) TBP:Pol II and TBP:FRB-tagged GTF occupancy ratios at the induced *HSP12* promoter in cells

*Figure 4 continued on next page*

*Figure 4 continued*
pretreated or not with rapamycin to deplete the indicated factors. (**D**) Pol II and TFIIB occupancies at the indicated promoters previously reported to have partial PICs (*Zanton and Pugh, 2006*) under normal (blue) and heat shock induction (red).
DOI: https://doi.org/10.7554/eLife.43654.007
The following figure supplement is available for figure 4:

**Figure supplement 1.** All GTFs are required for TBP occupancy, and hence PIC stability/formation.
DOI: https://doi.org/10.7554/eLife.43654.008

levels are extremely low after 4 hr. Interestingly, the average TBP:Pol II occupancy ratio near the promoter decreases at intermediate times (15–60 min) of uracil depletion (*Figure 5D* and *Figure 5—figure supplement 1D*), as would be expected from Pol II buildup caused by reduced elongation due to decreased (but non-zero) UTP levels. At the heat shock genes, TBP and Pol II occupancies sharply increase at early times, presumably due to the stress response to uracil depletion, but they drop to virtually undetectable levels after 2–4 hr. In addition, TBP occupancy at Pol III and Pol I promoters is also extremely low upon uracil depletion.

The drastic drop in PIC levels upon uracil removal is unlikely to be due to growth arrest per se, because depletion of Kin28 (*Wong et al., 2014*) or Taf1 (see below) only modestly reduces TBP and Pol II occupancy, even though cell growth is blocked. To address whether PIC instability is due to metabolic limitation per se, we performed a similar experiment depleting cells of leucine (the strain is also a *leu2* auxotroph). Leucine-depleted cells show a similar growth pattern as uracil-depleted cells with growth cessation at 4 hr. In contrast to the uracil-depleted cells, leucine-depleted cells do not show a drop in TBP and Pol II occupancy at all genes, although growth-inhibited genes are affected (*Figure 5A–C*, *Figure 5—figure supplement 1*). In addition, leucine-depleted cells do not show early induction of PIC levels at heat shock genes, presumably because they do not undergo the same stress response, nor do they show increased Pol II:TBP ratios at intermediate times of leucine depletion (*Figure 5D* and *Figure 5—figure supplement 1D*). The differences between uracil- and leucine-depleted cells are consistent with transcriptional profiling experiments in auxotrophic cells grown in chemostats at various concentrations of the required metabolites (*Brauer et al., 2008*). Thus, our results suggest that the drastic and general drop in PIC levels upon uracil removal is due to the absence of UTP precursors.

## Depletion of FACT strongly reduces PIC levels

As depletion of uracil, and consequently UTP, blocks transcriptional elongation, we considered the possibility that other factors involved in the elongation process might also affect PIC levels. We therefore analyzed PIC levels and transcription upon depletion of FACT, a histone chaperone complex that travels with elongating Pol II in vivo (*Mason and Struhl, 2003*) and is important for elongation through chromatin templates in vitro (*Orphanides et al., 1998*). FACT does not directly associate with promoters, but rather associates (directly or indirectly) with the elongation machinery after Pol II escapes from the promoter (*Mason and Struhl, 2003*). In accord with previous results (*Pathak et al., 2018*), depletion of FACT strongly reduces Pol II occupancy throughout the genome (*Figure 6*). More interestingly, TFIIB occupancy at essentially all promoters is reduced to a comparable extent, indicating that FACT is important for PIC formation or stability. As FACT is not a component of the PIC, these observations suggest that some aspect of FACT-dependent elongation is important for PIC levels. As such, these observations are consistent with the drastic decrease in PIC levels upon uracil depletion, but the mechanism by which FACT affects PIC levels is unknown.

## Taf1 is selectively important, but not required for transcription

The role of TFIID is controversial with respect to whether it is selectively (*Moqtaderi et al., 1996*; *Kuras et al., 2000*; *Li et al., 2000*; *Basehoar et al., 2004*) or generally (*Warfield et al., 2017*) required for transcription. Genome-scale analysis of Pol II occupancy in TAF1-depleted cells grown in SC medium reveals an overall ~2 fold decreased in transcription (*Figure 2A,B*). Furthermore, transcription is significantly more affected at TATA-lacking vs. TATA-containing genes as observed by metagene (*Figure 7A*) or individual gene analyses (*Figure 7B*).

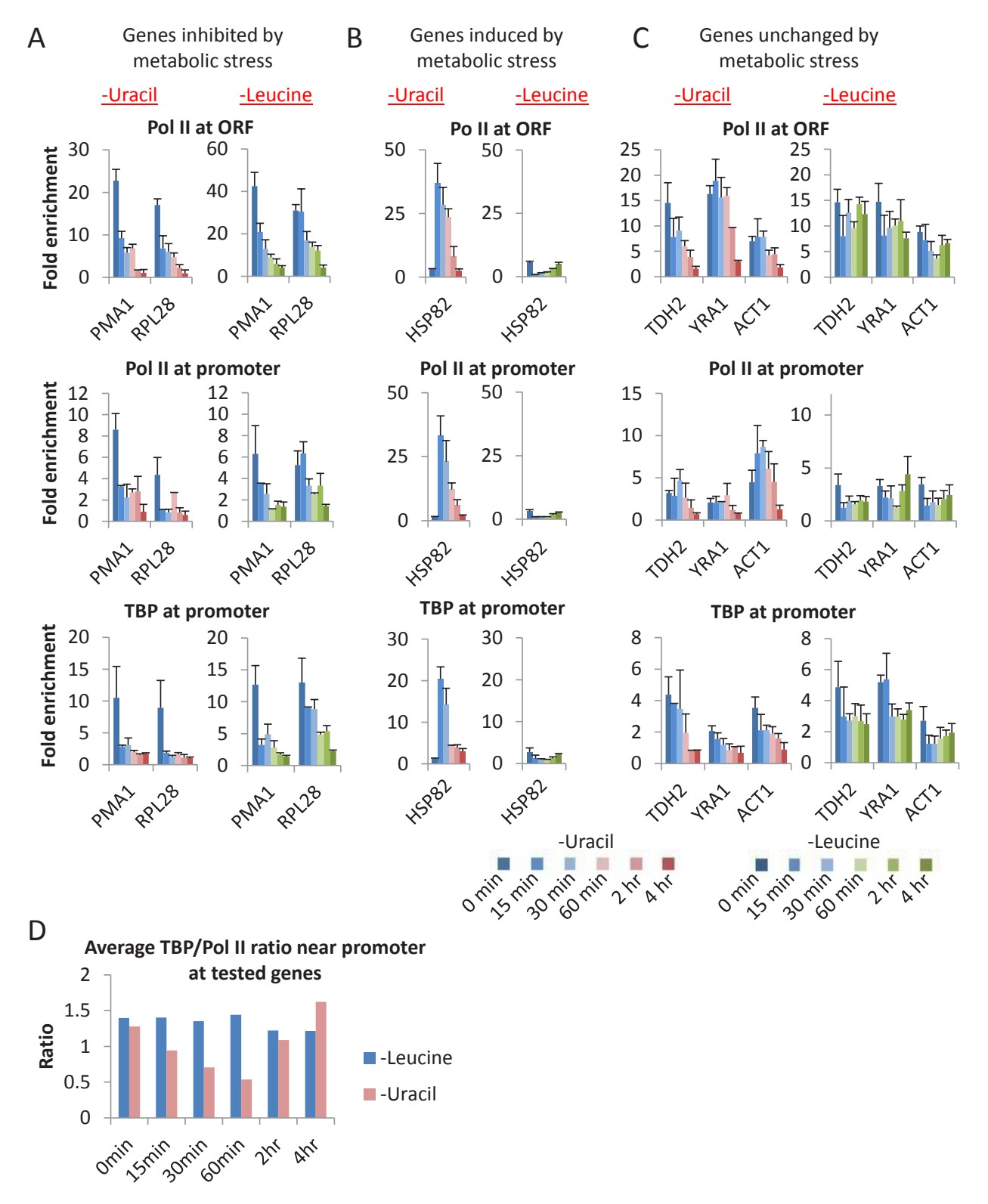

**Figure 5.** A general loss of PICs in cells depleted for uracil. (**A**) Pol II (ORF and promoter) and TBP occupancies at promoters (*PMA1* and *RPL28*) of genes inhibited by metabolic stress in cells depleted for uracil or leucine for various times (color scale). (**B**) Similar analysis for *HSP82*, a gene induced by various metabolic stresses. (**C**) Similar analysis for genes *TDH2*, *YRA1*, and *ACT1*, genes unchanged upon metabolic stress. (**D**) Average TBP/Pol II

*Figure 5 continued on next page*

*Figure 5 continued*

ratio at all tested promoters at various times after depletion of leucine (blue) or uracil (red). See *Figure 5—figure supplement 1D* for individual tested promoters.

DOI: https://doi.org/10.7554/eLife.43654.009

The following figure supplement is available for figure 5:

**Figure supplement 1.** A general loss of PICs in cells depleted for uracil.

DOI: https://doi.org/10.7554/eLife.43654.010

To exclude the possibility that these modest transcriptional effects are due to incomplete TAF1 depletion, we analyzed TBP, TFIIA, TFIIB (*Figure 7C*) and Taf1 (*Figure 7D*) occupancies at several promoters. Importantly, when Taf1 is depleted and no longer detected, TBP, TFIIA, and TFIIB strongly associate with promoters, but the effects depend on whether the promoter is 'SAGA- or TFIID-dependent'. At 'SAGA-dependent' promoters (*PMA1*, *CCW12*), Taf1 levels are low, and Taf1 depletion has marginal effects on GTF occupancy. As a consequence, the TBP:Taf1 occupancy ratio at these promoters are far above those occurring in non-depleted cells (*Figure 7E*). In contrast, at 'TFIID-dependent' promoters with relatively high levels of Taf1 occupancy (*RPL28* and *RPS14B*), GTF occupancy decreases upon Taf1 depletion, and the TBP:Taf1 occupancy ratio is only slightly affected (*Figure 7E*). In contrast and as shown above (*Figure 2C*), depletion of any individual GTF does not significantly alter the GTF:TBP or GTF:Pol II ratio. As expected, the TBP:TFIIA and TBP:TFIIB occupancy rations are similar in non-depleted and TAF1-depleted cells (*Figure 7—figure supplement 1A*). Thus, TAF1 behaves differently than all GTFs and hence is selectively important but not required for Pol II transcription.

Our results are in apparent contrast to those of a recent study that claimed that Taf-depletion (via auxin-induced degradation) resulted in 'similar transcription decreases for genes in the Taf-depleted, Taf-enriched, TATA-containing, and TATA-less gene classes' (*Warfield et al., 2017*). However, the conclusion of this other study was based on analysis of all ~5000 yeast genes, most of which are transcribed at low or even background levels that prevent accurate measurements. We performed a similar analysis of all ~5000 genes using the Taf1-depletion data presented here and confirmed a very modest difference between SAGA and TFIID-dependent promoters (*Figure 7F*; see 'all SAGA' and 'all TFIID'). Our analysis involves Pol II occupancy between +200 to+400 from the transcription start site, which avoids complications between transcriptional initiation and elongation. Although (*Warfield et al., 2017*) analyzed the region between 0 and +100, analysis of our Taf1-depletion data

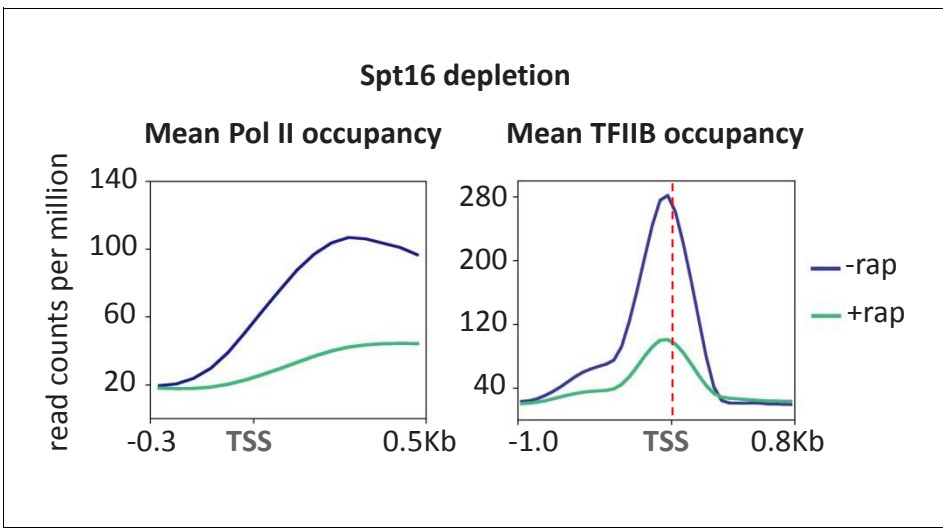

**Figure 6.** Depletion of Spt16 subunit of FACT reduces transcription and PIC formation. Mean Pol II occupancy and TFIIB occupancy averaged over 453 well-transcribed genes and promoters before and after Spt16 depletion.

DOI: https://doi.org/10.7554/eLife.43654.011

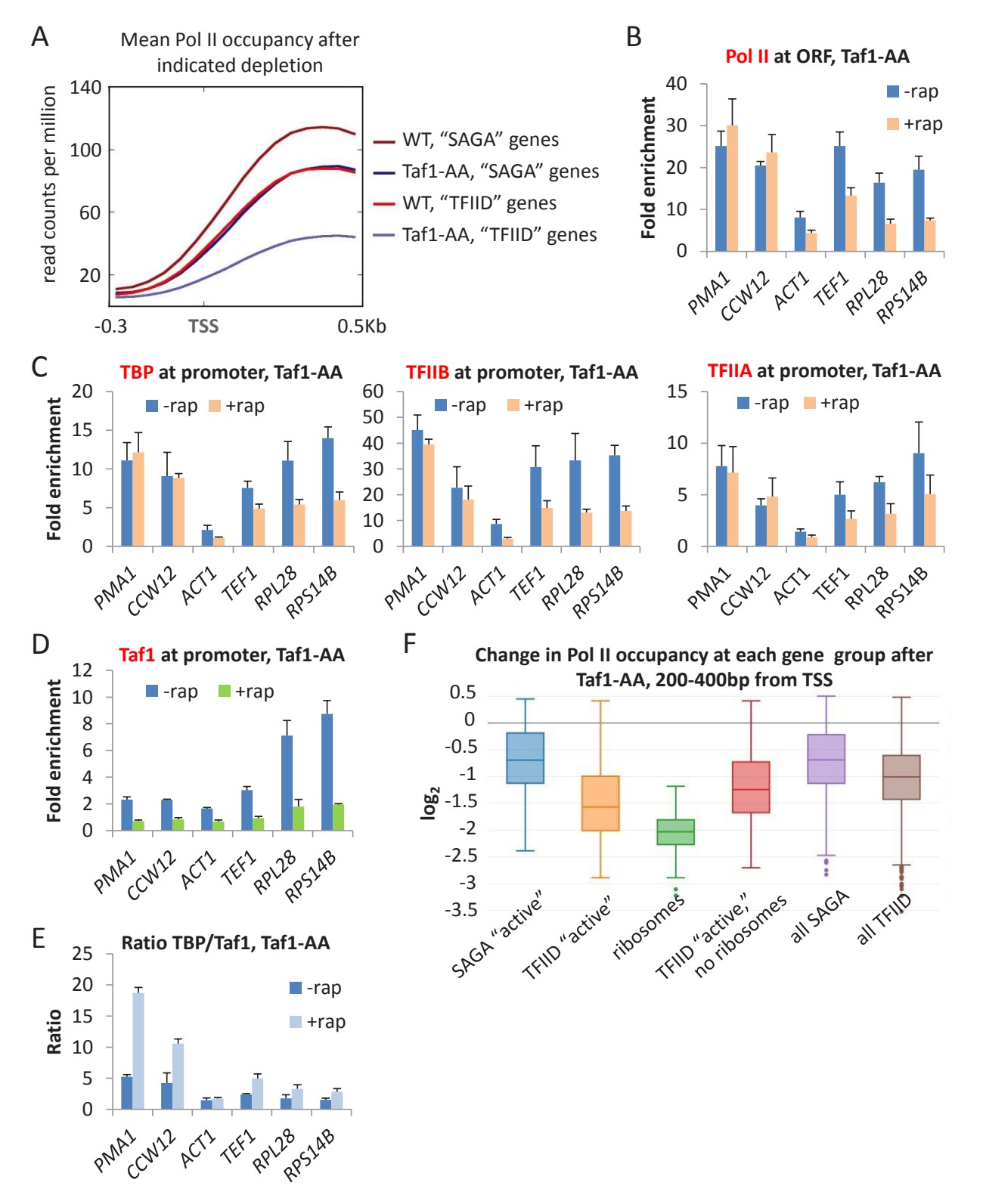

**Figure 7.** Taf1 depletion selectively affects TFIID-dependent genes. (**A**) Mean Pol II occupancy averaged over 453 well-transcribed 'SAGA' or 'TFIID' genes in parental or Taf1-depleted strains. (**B**) Pol II (**C**) TBP, TFIIB, and TFIIA, and (**D**) Taf1 occupancies at 'SAGA' (*PMA1* and *CCW12*) or 'TFIID' (*ACT1*, *TEF1*, *RPL28*, *RPS14B*) coding regions in the Taf1-depletion strain treated or untreated with rapamycin. (**E**) TBP:TAF1 occupancy ratios in the Taf1-depletion strain treated or untreated with rapamycin. (**F**) Log₂ change in Pol II occupancy (measured at +200–400 from the TSS) for the indicated gene

*Figure 7 continued on next page*

*Figure 7 continued*

classes upon Taf1 depletion. 'Active' genes are the top 10% of transcribed genes, broken up into 'SAGA' and 'TFIID'-dependent categories. Ribosomes include only the ribosomal genes from the TFIID-dependent category. 'TFIID active, no ribosomes' are the remainder of the top 10% transcribed genes in the TFIID-dependent category after ribosomal genes have been removed. 'All' genes are the entire set of ~5000 *s. cerevisiae* genes, broken up into 'SAGA' and 'TFIID'-dependent categories.

DOI: https://doi.org/10.7554/eLife.43654.012

The following figure supplements are available for figure 7:

**Figure supplement 1.** Taf1 depletion selectively affects TFIID-dependent genes.

DOI: https://doi.org/10.7554/eLife.43654.013

**Figure supplement 2.** Analysis of Taf11 and Taf13 depletion in SC medium using published data (*Warfield et al., 2017*) shows selective effects at TFIID-dependent genes.

DOI: https://doi.org/10.7554/eLife.43654.014

in this region also shows that the distinction between 'all SAGA' and 'all TFIID' genes is virtually non-existent (*Figure 7—figure supplement 1B*).

When we restrict the analysis to more actively transcribed genes, for which Pol II occupancy levels are well above the background (i.e. the top 453 as shown in *Figure 2A,B* and *Figure 7A*), the distinction between 'active' SAGA and TFIID genes is clear (*Figure 7F*). In addition, among TFIID-affected genes, ribosomal protein genes are even more strongly affected by Taf1 depletion (*Figure 7F*). In accord with this result, analysis of the same set of actively transcribed genes upon auxin-mediated depletion of Taf11 or Taf13- (*Warfield et al., 2017*) shows a clear distinction between SAGA and TFIID genes (*Figure 7—figure supplement 2*) and comparable to the Taf1-depletion data presented here. Thus, our experimental results are completely consistent with those of *Warfield et al. (2017)*, and they clearly demonstrate that TFIID-specific Tafs are selectively important for transcription of TATA-less genes.

## Discussion

### All GTFs including TFIIA are generally required for pol II transcription in vivo

Although GTFs were defined originally as factors required for 'basal' transcription in vitro, some GTFs are dispensable or only stimulatory under certain reaction conditions. In vivo, only some GTFs have been examined for their requirement for Pol II transcription, and few of these studies were performed on a whole-genome scale. Here, we show that every GTF is essential for Pol II transcription of essentially all genes. The limited amount of Pol II transcription observed upon GTF depletion is due to incomplete depletion, not GTF-independent transcription. While the basic observation is perhaps to be expected, the discrepancies between the in vitro and in vivo results are noteworthy. For example, TFIIE, TFIIF, and TFIIH are not required for in vitro transcription on negatively supercoiled templates (*Parvin and Sharp, 1993*; *Goodrich and Tjian, 1994*; *Parvin et al., 1994*; *Timmers, 1994*). While mechanistically interesting for understanding the transcription process, this observation is not relevant under physiological conditions, and it suggests that the DNA template is not negatively supercoiled in vivo.

The general requirement for TFIIA is unexpected. Under many in vitro conditions that involve normal DNA templates, TFIIA is stimulatory but not absolutely required (*Ozer et al., 1994*; *Sun et al., 1994*; *Yokomori et al., 1994*). In apparent agreement, multiple studies in vivo concluded that depletion of TFIIA generally reduces, but does not eliminate, transcription (*Chou et al., 1999*; *Liu et al., 1999*; *Stargell et al., 2000*). However, these in vivo experiments had two problems that could not be addressed at the time, because the method of chromatin immunoprecipitation had just started being applied to transcription factors. First, these experiments involved mRNA measurements, which are not a direct assay of transcription, especially given the then unknown fact that transcription and mRNA decay are coupled (*Haimovich et al., 2013*). Second, there was no way to assess whether the modest transcriptional effects of TFIIA depletion reflected incomplete depletion. By directly analyzing transcription and determining the TFIIA:TBP:Pol II occupancy ratios under TFIIA-depletion conditions, we demonstrate that TFIIA behaves indistinguishably from other GTFs in vivo. This indicates

that the ability of TFIIA to stabilize TBP binding to promoters is essential under physiological conditions.

Results presented here (see below) and elsewhere (*Petrenko et al., 2017*) indicate that although TAFs and Mediator contribute to PIC levels at most (and probably all) promoters, they are not essential for PIC formation and hence are not GTFs. In contrast, related studies using auxin-induced degradation to deplete these factors claimed that these factors are essential and hence comparable to GTFs (*Warfield et al., 2017*). However, our analysis of *Warfield et al. (2017)* data yield PIC and transcription levels that are comparable to our results using the anchor-away method for depletion of these factors (*Figure 7—figure supplement 2A*). While we suspect that *Warfield et al. (2017)* interpreted the reduced PIC and transcription levels as a consequence of incomplete depletion, our direct analyses of factor occupancy here (*Figure 7*) and elsewhere (*Petrenko et al., 2017*) clearly indicate that transcriptionally active PICs can exist in the absence of TAFs or Mediator but not GTFs.

## Partial and transcriptionally inactive PICs are extremely unstable in vivo

In vitro, the full complement of GTFs forms a PIC that is very stable in the absence of nucleotide precursors. Moreover, stepwise addition of GTFs results in a series of stable, partial PICs (*Buratowski et al., 1989*). In contrast, TBP occupancy is drastically reduced upon depletion of any individual GTF, indicating that partial PICs exist at very low levels in vivo. In addition, PICs are virtually non-detectable when cells are depleted of uracil, presumably due to a transcriptionally inactive PIC. Consistent with this, the relative ratios of GTF occupancies are similar on a genome-wide level (*Rhee and Pugh, 2012*), indicating that PICs are compositionally identical under the condition tested. As such, cells have a mechanism(s) to prevent stable, transcriptionally inactive PICs despite the intrinsic ability of the GTFs to form them.

Our conclusion about the instability of partial PICs is in apparent conflict with two previous observations. First, it has been claimed that partial PICs exist at some promoters during a mild heat shock (*Zanton and Pugh, 2006*). However, our re-analysis of the underlying data as well as direct experimental evidence questions the validity of the claim. Second, TBP can remain at TAF-dependent promoters upon thermal inactivation of TFIIB or the Med17 subunit of Mediator (*Li et al., 2000*). There are several possible explanations, not mutually exclusive, for this apparent discrepancy. The ts mutants used in these earlier experiments may not (and are unlikely to) completely inactivate TFIIB or Mediator, and promoter occupancy of these factors upon thermal inactivation was not assessed. In addition, as transcription measurements relied on mRNA levels only 45 min after the ts shift, considerable mRNA remained from before the shift (i.e. wasn't degraded), thereby making it impossible to measure modest levels of transcription. Indeed, more recent experiments using Pol II occupancy as an assay indicate that substantial transcription occurs in the Med17 ts (and anchor-away) strains, and transcription and TBP occupancy is more efficient at TAF-dependent promoters (*Petrenko et al., 2017*). However, as the Pol II occupancy measurements here are made 1 hr after GTF depletion, we cannot exclude the possibility that metastable partial PICs might exist at earlier time points.

In wild-type cells, Mediator is part of the PIC, but it dissociates from the PIC upon TFIIH-dependent phosphorylation of the C-terminal tail of Pol II, thereby permitting rapid promoter escape (*Jeronimo and Robert, 2014*; *Wong et al., 2014*). Mediator dissociation from the PIC is rapid (estimated life-time 1/8 s), and it leaves behind a post-escape complex containing GTFs that is capable of re-initiation with a new Pol II molecule (*Wong et al., 2014*). A Mediator-lacking PIC is competent for transcription in vivo, albeit at a lower level than a wild-type PIC (*Petrenko et al., 2017*). Thus, the PIC is extremely dynamic, both with respect to Mediator and to the instability of partial PICs.

The lack of PICs in uracil-depleted cells does not simply reflect cessation of cell growth or metabolic mayhem, because considerable PIC levels are observed upon depletion of Kin28, TAF1, or leucine, conditions in which cell growth is blocked. The observation that general PIC loss is observed specifically during uracil depletion suggests that the effect is due to the elimination of UTP precursors. In support of this conclusion, the Pol II:TBP occupancy ratio increases at intermediate levels of uracil depletion (intermediate time points), presumably reflecting poor elongation under UTP-limiting conditions. Taken together, these observations strongly suggest that, unlike the situation in vitro, a transcriptionally inactive PIC is unstable in vivo.

In principle, UTP limitation could affect PIC formation by leading to degradation, nuclear export, or an inactivating modification of a GTF. Alternatively, it could induce/activate a general inhibitory

factor that blocks PIC formation or clears PICs from the promoter. The fact that uracil depletion results in loss of Pol II and Pol III PICs suggests that the ultimate target of the signal might be something in common between these different transcription machineries, such as TBP or the common subunits of the RNA polymerases.

## TFIID generally contributes to and is differentially important for Pol II transcription, but it is not a required component of the PIC

Genetic and chromatin immunoprecipitation experiments over the past two decades indicate that the TAF subunits of TFIID are not generally required for transcription in yeast cells, but are selectively important at certain classes of promoters (*Struhl, 1986*; *Chen and Struhl, 1988*; *Struhl, 1987*; *Iyer and Struhl, 1995*; *Moqtaderi et al., 1996*; *Kuras et al., 2000*; *Li et al., 2002*; *Basehoar et al., 2004*; *Huisinga and Pugh, 2004*). However, these conclusions were questioned by a recent study (*Warfield et al., 2017*). As discussed below, we believe that the results here support the original conclusions about the role of TFIID in yeast cells.

As correctly noted (*Warfield et al., 2017*), early studies of Taf function involved measurements of mRNA levels, which do not necessarily reflect transcriptional activity, especially given the mechanistic connection between mRNA synthesis and mRNA decay (*Haimovich et al., 2013*). Upon analysis of newly synthesized mRNA, Taf1 depletion (via auxin-mediated proteolysis) in YPD medium caused a general 5-fold reduction in transcription (*Warfield et al., 2017*). However, Taf1 depletion in SC medium caused only a general 2-fold defect (*Warfield et al., 2017*), consistent with our results using anchor-away-mediated depletion in SC medium. This difference in Taf1 function in YPD vs. SC medium is reminiscent of the difference in Mediator recruitment by activator proteins under these two conditions. As activator function typically increases in sub-optimal growth conditions (*Fan et al., 2006*; *Fan and Struhl, 2009*), we suggest that the reduced activator function in YPD medium makes transcription more dependent on TFIID than SAGA. Importantly, in contrast to the dramatic decrease in transcription upon depletion of any individual GTF, the modest decrease in transcription in Taf1-depleted cells strongly argues against a general, GTF-like requirement for TFIID.

The long-standing belief that TFIID is selectively important at different classes of promoters was based on two complementary findings. First, unlike the constant GTF:GTF occupancy ratios at all promoters, the Taf1:GTF ratio is variable, strongly suggesting that there are TAF-containing and TAF-independent forms of transcriptionally active TBP (*Kuras et al., 2000*; *Li et al., 2000*). Formally, the different Taf1:GTF ratio could reflect differential crosslinking of TFIID at various promoters. However, our observation that substantial transcription and TBP occupancy occurs when Tafs are virtually absent from the promoter (i.e. a Taf1:TBP occupancy ratio of near zero) directly demonstrates the existence of a functional PIC lacking TAFs. Importantly, in wild-type cells, the Taf1:TBP occupancy ratios vary between 0.2 and 1.0, indicating that TAFs (and hence TFIID) contribute to transcription at all promoters (*Kuras et al., 2000*)(*Figure 7E*). Thus, the recent whole-genome analysis showing Taf occupancy at all promoters (*Warfield et al., 2017*) is consistent with previous results and does not indicate that TFIID is generally required for transcription.

Second, genes showing higher relative levels of Tafs (and hence TFIID) show stronger reductions in transcription upon Taf depletion, thereby linking TFIID occupancy with TFIID-transcription. Our whole-genome analysis extends this observation and clearly demonstrates that depletion of TAF1 affects 'TFIID-dependent' genes more strongly than 'SAGA-dependent' genes. In contrast, (*Warfield et al., 2017*) reported that Taf depletion does not differentially affect TFIID vs. SAGA genes. In resolving this apparent discrepancy, we note that our analysis was performed on 453 active genes, because Pol II occupancy at less expressed genes is at the detection limit. In contrast, (*Warfield et al., 2017*) performed their analysis on all genes, and we suspect that the poor signal: noise of the vast majority of genes masked the TFIID vs. SAGA distinction.

From all these considerations, we reiterate our long-standing view of TFIID function in yeast cells (*Kuras et al., 2000*). There are TAF-containing (TFIID) and Taf-lacking forms of transcriptionally active TBP. TFIID associates with promoters and contributes to transcription of all genes, but substantial transcription can occur in the absence of Tafs. The relative usage of the two TBP forms depends on the promoter, and occurs in a continuum. The TFIID form is particularly important at TATA-lacking promoters, reflecting the importance of TAF:DNA contacts in the absence of a canonical TATA element (*Verrijzer et al., 1994*; *Verrijzer et al., 1995*; *Oelgeschläger, et al., 1996*; *Burke and Kadonaga, 1997*). The Taf-independent form typically requires a canonical TATA

element for a stable interaction with TBP. Lastly, it appears that the relative usage of the two forms depends on the growth conditions, possible due to different levels of activator function which is linked to SAGA recruitment.

The key feature of this view involves the two different forms of TBP (Taf-containing and Taf-lacking) at the promoter. As both forms typically contribute (to various extents) to transcriptional activity of a given gene, the frequently used terms 'TFIID-dependent' and 'TFIID-independent' (typically also called 'SAGA-dependent') genes are misleading. Furthermore, while TATA elements are less common in 'TFIID-dependent' genes as compared to 'TFIID-independent' genes, TATA elements are not the sole determinant of the relative usage of the two forms of transcriptional active TBP. In this regard, TFIID is specifically recruited to ribosomal promoters by the RAP1-containing activator associated with the NuA4 histone acetylase co-activator complex (*Li et al., 2002*; *Mencía et al., 2002*; *Uprety et al., 2012*; *Uprety et al., 2015*). Thus, the mechanistic distinction refers to the differences in the basic transcriptional machinery, not the promoters or genes. The multiple yeast Pol II machineries active at individual promoters are conceptually analogous to the multiple bacterial RNA polymerases that differ by the σ factor bound to the core enzyme and that can act at many individual promoters (*Wade et al., 2006*).

## Materials and methods

### Yeast strain and growth conditions

Strains used in this study are listed in *Supplementary file 1*. Anchor-away strains were constructed as described previously (*Wong et al., 2014*). For spotting assays, yeast cells were grown at 30°C to an $OD_{600}$ of 0.3–0.5, diluted to 0.1, and 5-fold serial dilutions of cells were spotted on plates containing YPD medium with or without 1 μg/ml rapamycin for 48–60 hr. For all other experiments (except for that involving Sp16 depletion, which was performed in YPD medium), strains were grown in SC liquid media to an $OD_{600}$ of 0.4, and rapamycin was then added to a final concentration of 1 μg/ml for 1 hr except when otherwise indicated. For heat shock experiments, cells (pretreated or not with rapamycin for 45 min) were grown at 30°C, filtered, and then transferred to pre-warmed 39°C medium for 15 min in the presence or absence of rapamycin. For the uracil and leucine depletion experiments, cell grown in SC medium were filtered, washed with appropriate medium, and transferred to SC medium lacking either uracil or leucine for the indicated times.

### Chromatin immunoprecipitation (ChIP)

Chromatin, prepared as described previously (*Fan et al., 2008*) from 5 ml of cells ($OD_{600}$ ~0.5), was immunoprecipitated with antibodies against Pol II unphosphorylated CTD (8WG16, Covance), FRB (Enzo Life Sciences), Taf1 (a kind gift from Steve Buratowski), TBP, TFIIA, and TFIIB. Immunoprecipitated and input samples were analyzed by quantitative PCR in real time using primers for genomic regions of interest and a control region from chromosome V to generate IP:input ratios for each region. The level of protein association to a given genomic region was expressed as fold-enrichment over the control region. At least three biological replicates (culture samples collected on separate days, with lysis and IP performed on separate days) were performed for each experiment, and each sample was analyzed in triplicate by qPCR (technical replicates) to obtain an average value for that sample. Error bars represent the standard deviation between the biological replicates.

### ChIP-seq and data analyses

Barcoded sequencing libraries from ChIP DNA (two biological replicates per strain) were constructed as described previously (*Wong et al., 2013*). Sequence reads were mapped using Bowtie available through the Galaxy server (Penn State) with the following options: *-n 2, -e 70, -l 28, -v −1, -k 1, -m −1*. Normalization was performed relative to *S.pombe* DNA, added in equal amounts to each sample before immunoprecipitation, for the replicates with spike-in, and relative to the number of mapped reads, as well as the median Pol II levels at a non-transcribed region of Chromosome V set as the 'background level' for the replicates without spike-in; similar results were obtained with and without spike-in. ChIP-seq data were visualized using the Integrated Genome Browser. Mean occupancy curves were generated using Galaxy deepTools (Freiburg, Germany), scaled relative to the number of mapped reads and fragment size, and expressed as counts per million mapped reads

(CPM). Individual gene analysis for fold Pol II occupancy change was likewise performed using Galaxy deepTools. TFIID- and SAGA-dependent genes were defined previously (*Basehoar et al., 2004*; *Huisinga and Pugh, 2004*). Boxplots were generated using Plotly Chart Studio (https://plot.ly/create/box-plot/). The ChIP-sequencing data and associated files are available through the Gene Expression Omnibus (GEO) under the accession number GSE122734.

## Acknowledgements

We thank Steve Buratowski for antibodies against Taf1. This work was supported by a Croucher Foundation Fellowship and research grants MYRG2015-00186-FHS and MYRG2016-0-0211-FHS from the University of Macau to KHW and by a research grant to KS from the National Institutes of Health (GM30186).

## Additional information

### Competing interests

Kevin Struhl: Senior editor, *eLife*. The other authors declare that no competing interests exist.

### Funding

| Funder | Grant reference number | Author |
| --- | --- | --- |
| National Institutes of Health | GM 30186 | Natalia Petrenko<br>Yi Jin<br>Koon Ho Wong<br>Kevin Struhl |
| Universidade de Macau | MYRG2015-00186 FHS | Liguo Dong<br>Koon Ho Wong |
| University of Macau | MYRG2016-00211-FHS | Liguo Dong<br>Koon Ho Wong |

The funders had no role in study design, data collection and interpretation, or the decision to submit the work for publication.

### Author contributions

Natalia Petrenko, Yi Jin, Conceptualization, Resources, Data curation, Software, Formal analysis, Validation, Investigation, Visualization, Methodology, Writing—original draft, Writing—review and editing; Liguo Dong, Investigation; Koon Ho Wong, Conceptualization, Resources, Data curation, Software, Formal analysis, Supervision, Funding acquisition, Validation, Investigation, Visualization, Methodology, Writing—original draft, Writing—review and editing; Kevin Struhl, Conceptualization, Formal analysis, Supervision, Funding acquisition, Validation, Visualization, Writing—original draft, Project administration, Writing—review and editing

### Author ORCIDs

Koon Ho Wong https://orcid.org/0000-0002-9264-5118
Kevin Struhl http://orcid.org/0000-0002-4181-7856

### Decision letter and Author response

Decision letter https://doi.org/10.7554/eLife.43654.022
Author response https://doi.org/10.7554/eLife.43654.023

## Additional files

### Supplementary files

• Supplementary file 1. List of strains.
DOI: https://doi.org/10.7554/eLife.43654.015

• Transparent reporting form
DOI: https://doi.org/10.7554/eLife.43654.016

### Data availability

Sequencing data has been deposited in GEO under the accession number GSE122734.

The following dataset was generated:

| Author(s) | Year | Dataset title | Dataset URL | Database and Identifier |
|---|---|---|---|---|
| Petrenko N, Jin Y, Dong L | 2019 | Requirements for RNA polymerase II preinitiation complex formation in vivo | https://www.ncbi.nlm.nih.gov/geo/query/acc.cgi?acc=GSE122734 | NCBI Gene Expression Omnibus, GSE122734 |

The following previously published dataset was used:

| Author(s) | Year | Dataset title | Dataset URL | Database and Identifier |
|---|---|---|---|---|
| Warfield L, Rama-chandran S, Hahn S | 2017 | Transcription of nearly all yeast RNA Polymerase II-transcribed genes is dependent on transcription factor TFIID | https://www.ncbi.nlm.nih.gov/geo/query/acc.cgi?acc=GSE97081 | NCBI Gene Expression Omnibus, GSE97081 |

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
