## [Decision Letter]

Thank you for submitting your article "Requirements for preinitiation complex formation in vivo" for consideration by *eLife*. Your article has been reviewed by two peer reviewers, and the evaluation has been overseen by Michael Green as the Reviewing Editor and James Manley as the Senior Editor. The reviewers have opted to remain anonymous.

The reviewers have discussed the reviews with one another and the Reviewing Editor has drafted this decision to help you prepare a revised submission.

This is a well-written manuscript with compelling rationale for this study. Experiments are nicely designed with appropriate controls. Data are of high quality with statistical analysis, using biological and technical replicates. Based on their data, the authors conclude here that all GTFs (general transcription factors) are required for PIC (pre-initiation complex) formation in vivo, and suggest that partial PIC is not stable in vivo, in contrast to previous in vitro studies. Further, they conclude that PIC is not stable when transcription elongation does not occur. They also conclude in this manuscript that TAFs (TBP-associated factors) contribute to transcription of most genes. These are important observations, and will have significant impact in the field of transcription.

Please address all of the reviewers' comments below to the best of your ability.

Major substantive comments:

1) TAFs contribute to transcription of most genes.

This conclusion is solely based on deletion (anchor-away) of only TAF1, a specific component of TFIID. Inclusion of two other TFIID-specific TAFs would have significant impact on this conclusion.

2) PIC is not stable when transcription elongation does not occur.

This conclusion is primarily based on the analyses of TBP and RNA polymerase II occupancies at the promoter following uracil depletion that impairs transcription elongation. Performing similar ChIP experiments following genetic inhibition of a transcription elongation factor and/or pharmacological inhibition of transcriptional elongation would have significant impact on this conclusion.

3) Partial PIC is not stable in vivo.

This conclusion/suggestion is based on only TBP occupancy at the promoter in deletion backgrounds (anchor-away) of GTFs. Other components of the PIC are not analyzed here. Although TBP nucleates the PIC formation and other GTFs are not likely to be present at the promoter in the absence of TBP, it would be good to show that other components of PIC are not present at the promoter in the deletion backgrounds of various GTFs, thus supporting that partial PIC is not detectable by the ChIP assay. Alternatively and/or in addition, the authors may consider to pull down minimal core promoter following in vivo crosslinking and analyze what are the PIC components bound to the minimal core promoter with or without deletion of GTFs (or some cytological experiments using fluorescence microscopy may also help), though these experiments could be technically challenging.

Comments related to the text:

1) Previous in vivo studies from the Green laboratory have demonstrated that TFIIB and Mediator are dispensable for TBP recruitment at the TAF-dependent promoter, but are essential for TBP recruitment at the TAF-independent promoter (Li et al., 2000; Bhaumik et al., 2004). Mediator and TFIIB are required for transcription of both classes of genes (Li et al., 2000; Bhaumik et al., 2004). These studies indicate the existence of partial PIC at the promoter in vivo, as TBP remains bound to the TAF-dependent, but not TAF-independent, promoter in the TFIIB and Mediator mutants (Li et al., 2000; Bhaumik et al., 2004). In contrast, authors in this manuscript find that TFIIB depletion impairs TBP recruitment at all genes, and thus concludes that partial PIC is not stable in vivo. The authors need to take into consideration of these previous studies/facts (Li et al., 2000; Bhaumik et al., 2004) in their Results and Discussion in this manuscript.

2) Previous in vivo studies from the Struhl and Green laboratories (Mencía et al., 2002; Li et al., 2002) demonstrated that UAS or co-activator specificity of the activator predominantly contributes to the TAF-dependency or -independency, but not TATA-containing or lacking sequence. This was not apparent in this manuscript. The authors need to discuss this fact in the manuscript, especially in the last paragraph of the Discussion section.

3) The authors have written last but one paragraph in the Introduction section to make their case compelling for this study without discussing/mentioning several important facts (or publications) which are relevant here. For example, TAF-dependency or independency was published back to back in the same issue of Science by the Struhl and Green laboratories in 2000. Struhl publication is cited, but Green publication is not cited in the first sentence in this paragraph and several other places in this manuscript. In the second sentence of this paragraph, several important references from the Green and Struhl laboratories regarding TAF or SAGA dependency/independency are missing. These are: Mencía et al., 2002; Li et al., 2002; Bhaumik and Green 2001; and Bhaumik and Green 2002. The last sentence in this paragraph mentions that "However, this view has been challenged by experiments claiming that the two classes of promoters behave similarly upon TFIID depletion (Warfield et al., 2017)". However, the authors have not considered some important facts in line with this sentence. For example, previous studies demonstrated that NuA4 is required for TFIID recruitment at the ribosomal protein gene promoters (e.g., Reid et al., Molecular Cell, 2000; Uprety et al., 2012), and the absence of NuA4 impairs recruitment of TFIID to the ribosomal protein gene promoters (Uprety et al., 2015). Intriguingly, TBP (TAF-independent form of TBP) is recruited to the ribosomal protein gene promoters in the absence of NuA4 or TFIID, and such recruitment of TBP is mediated via SAGA (Uprety et al., 2015). Further, another recent study (Ferdoush et al., 2018) shows that SAGA is recruited to the PHO84 promoter for TFIID-independent transcription initiation in the YPD medium (or plus Pi). This gene can become TFIID-dependent in absence of Pi in the growth medium (Ferdoush et al., 2018). These studies indicate that the absence of TFIID can make a gene SAGA dependent and vice versa. These relevant studies may be mentioned here in this paragraph (or may be discussed later).

4) In the last paragraph of the Results section, the authors mention that "We also note that our analysis involves Pol II occupancy measurements at the region between +200 to +400 from the transcription start site, which avoids complications between transcriptional initiation and elongation, whereas the other study utilized the region between 0 and +100 (Warfield et al., 2017). Indeed, the distinction between "all TATA" and "all TFIID" genes is virtually eliminated when our Taf1-depletion data is analyzed between 0 and +100 (Figure 6—figure supplement 1B).". How does the change in position of Pol II analysis by about ~200 bp alter the conclusion? This needs to be appropriately discussed. How is the complication avoided between transcriptional initiation and elongation at the region between +200 and +400 from transcription start site? RNA polymerase II pausing could be an issue in this region.

---

## [Author Response]

Major substantive comments:1) TAFs contribute to transcription of most genes.This conclusion is solely based on deletion (anchor-away) of only TAF1, a specific component of TFIID. Inclusion of two other TFIID-specific TAFs would have significant impact on this conclusion.

Analyzing additional TAFs by anchor-away has minimal value given the point of the paper. Many TAF-inactivation experiments indicating that TFIID-specific TAFs selectively affect transcription. Regarding TAFs, the major point of our paper is to invalidate the opposite claim of Warfield et al., 2017. We do this by measuring TAF:GTF occupancies upon depletion of TAF1 (not done previously for any TAF-depletion experiment) and by re-analyzing the Warfield et al. genome-wide experiment (i.e. by restricting the analysis to genes whose expression is above the background). We also emphasize and discuss an interesting difference (discovered but largely ignored by Warfield et al.) with respect to the importance of TFIID in YPD vs. SC medium. Nevertheless, while not doing any new experiments ourselves, we did address this comment by analyzing Warfield et al. data on two other TFIID-specific TAFs (TAF11 and TAF13). The results (Figure 7—figure supplement 2) clearly demonstrate that 3 different TFIID-specific TAFs behave selectively and similarly with respect to the TFIID-SAGA distinction.

2) PIC is not stable when transcription elongation does not occur.This conclusion is primarily based on the analyses of TBP and RNA polymerase II occupancies at the promoter following uracil depletion that impairs transcription elongation. Performing similar ChIP experiments following genetic inhibition of a transcription elongation factor and/or pharmacological inhibition of transcriptional elongation would have significant impact on this conclusion.

The suggestion of inhibiting elongation sounds good, but there is no known factor or drug that specifically inhibits elongation in yeast cells. More importantly, the point of the uracil experiment is to recreate the classic biochemical definition of the PIC (i.e. absence of NTPs) in vivoand assess its stability in vivo. There is no criticism of the experiment or the conclusions that arise. However, to address this suggestion, we now include an experiment done long ago in which we analyzed PIC and transcription levels in FACT-depleted cells. FACT is a histone chaperone that travels with elongating Pol II, is important for elongation through chromatin templates in vitro, and does not associate with promoters. Interestingly, as shown in new Figure 6, FACT depletion strongly reduces Pol II transcription genome-wide (already published by Pathak et al., 2018), and TFIIB occupancy at promoters. We recognize that the molecular basis of this observation is unknown, but we do think it is an interesting connection between elongation and PIC formation/stability worth reporting. We are not willing to do more experiments on FACT, so if our current Figure 6 is not acceptable to the reviewers, we will simply remove it. As mentioned above, the FACT experiment is not directly relevant for the uracil-depletion experiment, which is a key conclusion of the paper.

3) Partial PIC is not stable in vivo.This conclusion/suggestion is based on only TBP occupancy at the promoter in deletion backgrounds (anchor-away) of GTFs. Other components of the PIC are not analyzed here. Although TBP nucleates the PIC formation and other GTFs are not likely to be present at the promoter in the absence of TBP, it would be good to show that other components of PIC are not present at the promoter in the deletion backgrounds of various GTFs, thus supporting that partial PIC is not detectable by the ChIP assay. Alternatively and/or in addition, the authors may consider to pull down minimal core promoter following in vivo crosslinking and analyze what are the PIC components bound to the minimal core promoter with or without deletion of GTFs (or some cytological experiments using fluorescence microscopy may also help), though these experiments could be technically challenging.

The suggestion to look at other GTFs upon depletion of an individual GTF is unnecessary and would take a vast amount of work. First of all, our experiments already involve 2 GTFs, TBP and Pol II. Second, numerous biochemical experiments and high-resolution structural knowledge make it virtually impossible that any GTF or combination of GTFs can stably interact with the promoter in the absence of TBP. Moreover, to do this suggestion properly, we would have to do a vast number of genome-wide experiments in which we look at every GTF for every individual GTF depletion. Otherwise, one could always say that upon depletion of factor X, factor Y remains associated with a subset of promoters.

Comments related to the text:1) Previous in vivo studies from the Green laboratory have demonstrated that TFIIB and Mediator are dispensable for TBP recruitment at the TAF-dependent promoter, but are essential for TBP recruitment at the TAF-independent promoter (Li et al., 2000; Bhaumik et al., 2004). Mediator and TFIIB are required for transcription of both classes of genes (Li et al., 2000; Bhaumik et al., 2004). These studies indicate the existence of partial PIC at the promoter in vivo, as TBP remains bound to the TAF-dependent, but not TAF-independent, promoter in the TFIIB and Mediator mutants (Li et al., 2000; Bhaumik et al., 2004). In contrast, authors in this manuscript find that TFIIB depletion impairs TBP recruitment at all genes, and thus concludes that partial PIC is not stable in vivo. The authors need to take into consideration of these previous studies/facts (Li et al., 2000; Bhaumik et al., 2004) in their Results and Discussion in this manuscript.

The reviewer correctly notes that previous results that TBP can remain at TAF-dependent promoters upon thermal inactivation of TFIIB or Mediator (Med17 subunit) are in apparent conflict with our conclusion that partial PICs do not exist at appreciable levels in vivo. There are several possible explanations, not mutually exclusive, for this apparent discrepancy. First, the ts mutants used in the earlier experiments may not (and are unlikely to) completely inactivate TFIIB or Mediator. Because these experiments were done so long ago, ChIP was not done to assess the promoter occupancy of these factors upon thermal inactivation, and the transcriptional results relied on mRNA levels that are subject to mRNA stability issues. Furthermore, as these mRNA measurements were made only 45 minutes after the ts shift, considerable mRNA remained from before the shift (i.e. wasn’t degraded), thereby making it impossible to measure low/modest levels of transcription. In this regard, our more recent experiments using Pol II occupancy as an assay indicate that substantial transcription does occur in the Med17 ts mutant (and anchor-away allele), and transcription and TBP occupancy is more efficient at TAF-dependent promoters (Petrenko et al., 2017). Second, as the measurements in our current paper are made 1 hr after GTF depletion, it is possible that partial PICs might be metastable and exist at earlier time points (not so easy to distinguish true partial PICs from incomplete depletion). Thus, our conclusion that partial PICs do not exist at appreciable levels is correct. However, we now discuss these issues in the revised manuscript.

2) Previous in vivo studies from the Struhl and Green laboratories (Mencía et al., 2002; Li et al., 2002) demonstrated that UAS or co-activator specificity of the activator predominantly contributes to the TAF-dependency or -independency, but not TATA-containing or lacking sequence. This was not apparent in this manuscript. The authors need to discuss this fact in the manuscript, especially in the last paragraph of the Discussion section.

We now mention the activator-specificity of TFIID recruitment to ribosomal protein promoters and cite the relevant papers. We note that this fact is largely irrelevant to the conclusions of our paper, which are concerned with whether TAFs are generally required or selectively important for transcription.

3) The authors have written last but one paragraph in the Introduction section to make their case compelling for this study without discussing/mentioning several important facts (or publications) which are relevant here. For example, TAF-dependency or independency was published back to back in the same issue of Science by the Struhl and Green laboratories in 2000. Struhl publication is cited, but Green publication is not cited in the first sentence in this paragraph and several other places in this manuscript. In the second sentence of this paragraph, several important references from the Green and Struhl laboratories regarding TAF or SAGA dependency/independency are missing. These are: Mencía et al., 2002; Li et al., 2002; Bhaumik and Green 2001; and Bhaumik and Green 2002. The last sentence in this paragraph mentions that "However, this view has been challenged by experiments claiming that the two classes of promoters behave similarly upon TFIID depletion (Warfield et al., 2017)". However, the authors have not considered some important facts in line with this sentence. For example, previous studies demonstrated that NuA4 is required for TFIID recruitment at the ribosomal protein gene promoters (e.g., Reid et al., Molecular Cell, 2000; Uprety et al., 2012), and the absence of NuA4 impairs recruitment of TFIID to the ribosomal protein gene promoters (Uprety et al., 2015). Intriguingly, TBP (TAF-independent form of TBP) is recruited to the ribosomal protein gene promoters in the absence of NuA4 or TFIID, and such recruitment of TBP is mediated via SAGA (Uprety et al., 2015). Further, another recent study (Ferdoush et al., 2018) shows that SAGA is recruited to the PHO84 promoter for TFIID-independent transcription initiation in the YPD medium (or plus Pi). This gene can become TFIID-dependent in absence of Pi in the growth medium (Ferdoush et al., 2018). These studies indicate that the absence of TFIID can make a gene SAGA dependent and vice versa. These relevant studies may be mentioned here in this paragraph (or may be discussed later).

We have cited the papers mentioned in this comment. We note that these comments are related to the role of NuA4 at ribosomal protein promoters and the role of SAGA, subjects that are only peripherally related to the experiments and conclusions here.

4) In the last paragraph of the Results section, the authors mention that "We also note that our analysis involves Pol II occupancy measurements at the region between +200 to +400 from the transcription start site, which avoids complications between transcriptional initiation and elongation, whereas the other study utilized the region between 0 and +100 (Warfield et al., 2017). Indeed, the distinction between "all TATA" and "all TFIID" genes is virtually eliminated when our Taf1-depletion data is analyzed between 0 and +100 (Figure 6—figure supplement 1B).". How does the change in position of Pol II analysis by about ~200 bp alter the conclusion? This needs to be appropriately discussed. How is the complication avoided between transcriptional initiation and elongation at the region between +200 and +400 from transcription start site? RNA polymerase II pausing could be an issue in this region.

The difference between Pol II occupancy measurements at +200- to +400 vs. those at 0 to +100 is very subtle. In both cases, the Warfield et al. analysis of “all genes” shows very little difference between TATA-containing and TATA-less promoters, whereas our analysis of the 500 most well-expressed genes (i.e. the ones where there is reliable data) shows considerable difference. We do not know why the measurements at these two different regions are subtly different and hence haven’t commented on this in the revised paper. Pol II pausing is not relevant here, because numerous genome-wide experiments indicate that pausing does not occur in yeast, unlike what occurs in many other eukaryotic organisms.